# Translating Multilingualism in Mira Nair's *Monsoon Wedding*

**Montse Corrius** [1,*], **Eva Espasa** [2] and **Laura Santamaria** [3]

1    Department of Communication, Faculty of Business and Communication, University of Vic—Central University of Catalonia, 08500 Vic, Spain

2    Department of Translation, Interpreting and Applied Languages, Faculty of Education, Translation, Sport and Psychology, University of Vic—Central University of Catalonia, 08500 Vic, Spain; eva.espasa@uvic.cat

3    Department of Translation, Interpretation and East Asia Studies, Faculty of Translation and Interpretation, Autonomous University of Barcelona, 08193 Barcelona, Spain; laura.santamaria@uab.cat

*    Correspondence: montse.corrius@uvic.cat

**Abstract:** Linguistic diversity is present in many audiovisual productions and has given rise to fruitful research on translation of multilingualism and language variation. *Monsoon Wedding* (Mira Nair, 2001) is a prototypical film for translation analysis, since multilingualism is a recurrent feature, as the film dialogue combines English (L1) with Hindi and Punjabi (L3), which creates an effect of code-switching. This article analyses how the multilingualism and the cultural elements present in the source text (ST) have been transferred to the Spanish translated text (TT) *La boda del monzón*. The results show that in the Spanish dubbed and subtitled versions, few Indian cultural elements are left, and little language variation is preserved. Thus, L3 does not play a central role as it does in the source text. In the translation, only a few loan words from Hindi or Punjabi are kept, mainly from the domains of food and cooking, as well as terms of address and greetings, or words related to the wedding ceremony. The results also show that when L3 is not fully rendered in translation, otherness is still conveyed through image and music, thus (re)creating a different atmosphere for Spanish audiences.

**Keywords:** third language; multilingualism; audiovisual translation; Bollywood; linguistic diversity; cultural elements

## 1. Multilingual Films: The Third Language

Although multilingual texts (written, oral, or audiovisual) have existed for centuries, the number of texts that use a combination of two or more languages seems to have increased in recent years, as our society is becoming more mixed and multicultural. Hence, linguistic diversity in audiovisual productions has gradually risen, which has resulted in a new approach to feature films by "incorporating the contemporary context of cultural exchange, characterised by cross-border flows of people, commodities and culture, into the story-world of the film" (UIS 2016, p. 4).

Linguistic diversity is a fundamental element of cultural divergence because, as Albornoz and García Leiva (2016) stated, "advocating linguistic diversity is an imperative for the international community, as every language reflects a one-of-a-kind vision of the world, with its own value system, its specific philosophy and its particular cultural charac-teristics". Similarly, Díaz-Cintas (2015) states that languages not only mark geographical and cultural borders, but also embody the characters' social, cultural, and personal features.

Cinema and audiovisual productions in general have always been a significant vehicle for the representation of nation and culture, and language has played a major role because it helps to locate characters' identities. Thus, the presence of different languages alerts viewers to a change of cultural setting, bearing in mind that the presence of linguistic diversity "applies not only to language and language difference but also to culture and identity" (Corrius and Zabalbeascoa 2019). This linguistic diversity, namely multilingualism, has

always been present in Hollywood productions (Bleichenbacher 2008a), but it was not until the 1980s and 1990s that the number of films requiring the audience to deal with communication in more than one language bloomed (Heiss 2004). As stated by Corrius and Zabalbeascoa (2011, p. 114), many popular English-language films display some other language(s) to a greater or lesser degree. These constitute instances of the kind of language variation coined by Corrius (2008) as the *third language* or *L3*. Hence, the third language is neither the main language used in the source text, nor the main language used in the target text, but a secondary language that exists in the ST and is also embodied in the process of translating. L3 may be a distinct, independent language or an instance of relevant language variation (Corrius and Zabalbeascoa 2011, p. 115). Following Corrius and Zabalbeascoa (2011, 2019) L3 is a concept of linguistic variation, which may include one language or more than one; the concept of third language might include more than one L3 (e.g., Hindi and Punjabi in *Monsoon Wedding*). "The concept of L3 stresses the fact that not all voices in the text (e.g., a film or a novel) speak the same language or the same variety" (Corrius and Zabalbeascoa 2011, p. 117).

*Monsoon Wedding* (2001) makes use of multilingualism and its nuances in India to give a realistic account of India. The presence of multilingualism in audiovisual productions in general, and in contemporary Hollywood films in particular, has been studied by a number of translation scholars, and in the funded research projects TRAFILM (2015–2018) and MUFiTAVi (2019–2021), and related publications. Furthermore, the use of a third language is also common in Bollywood films, since code-switching with English, which is understood as "the use of several languages or dialects in the same conversation or sentence by bilingual people" (Gardner-Chloros 2009, p. 4), is highly prevalent. This linguistic trait is a characteristic of Bollywood cinema (Bonsignori and Bruti 2014; Si 2011), along with some formal and aesthetic characteristics, such as the use of pervasive colours, the vivid and lurid forms, and the music and dance (Acciari 2013; Nair 2001). To put it in Sharma's (2014) words, Bollywood communicates "the Indian cultural diversities in a most creative manner with the efficient usage of colors, cultural signs and constructs across the borders".

Although it is not prototypically Bollywoodian, *Monsoon Wedding* (MW) helped the popularisation of Bollywood films in Europe (Acciari 2013), as it shows themes of love and arranged marriages, which are often included in Bollywood films. For Bowen (2002), it shares "the resplendent blur of music, dancing and pageant that is characteristic of Bollywood marriage films". As its director, Mira Nair, stated in an interview (Bowen 2002), it was the failure of traditional Bollywood films to represent modern contemporary India that initially inspired the film. MW is a portrait of contemporary India and, in Nair's words, "it is also a very personal story about my kind of people, Punjabi people. It is a community that works hard, parties hard and has a huge appetite for life" (Bowen 2002). According to Sharma (2014), MW is a depiction of Punjabi culture, which he defines as a popular "sub-culture" among the several Indian dominant cultures. Therefore, audiences are faced with many cultural references in the script and in the images conveying information about the scenery and the characters' identities. For Chiaro (2008, p. 156) cultural references are "entities that are typical of one particular culture, and that culture alone", such as words related to food, drinks, festivities, place names, names of famous people, objects, sports, units of measurement, etc. (Minutella 2012, p. 315).

In addition to being influenced and inspired by Bollywood cinema, MW has some other cinematic influences: the American documentary, the films of Emir Kusturica and Fellini; this is the reason why the last section of this paper (see Section 3.3 below) is dedicated to the importance of images. Indeed, it is an international co-production between India, the United States, and Europe (France, Germany, and Italy). It does not belong to the Hindi film industry based in Mumbai as Bollywood productions do but, as stated by Acciari (2013), it generated interest in the reading of Bollywood cinema as a transnational phenomenon. Hence, it can be considered an important element in studies of Bollywood cinema.

Within this context, this article examines (1) how the cultural elements and the third languages present in the source text, which helped Nair recreate the Indian culture, have

been transferred to the Spanish dubbed and subtitled versions, (2) what L3 strategies have been used in the translation, and (3) how the feeling of otherness remains in the target text (TT).

## 2. Materials and Methods

To carry out the study we analysed the source (English) and target texts (dubbed and subtitled versions) of *Monsoon Wedding* (2001), a comedy set in New Delhi by the Punjabi director Mira Nair. For this particular research, and following Bonsignori and Bruti (2014), who analysed the Italian dubbed version of *Monsoon Wedding* (MW), we examined in detail the domains of food and cooking, as well as terms of address and greetings, or words related to the wedding ceremony, as they are the terms in Hindi and Punjabi mostly kept in the TT. The film reveals the interplay between traditional Indian culture (arranged marriages) and Indian modernity (the proximity of upper and lower classes among modern young people and the mixture of languages). Members of a middle-class Indian family return to India from different parts of the world for a wedding. The bridegroom, Hemant, a computer analyst who lives in Houston, returns home to marry a woman (Aditi) who he has never met. Mr. Verma, Aditi's father, is making last-minute preparations for his daughter's big wedding.

The presence of L3 is a recurrent feature throughout the film; the linguistic variety takes place both through different languages (Indian English, Hindi, and Punjabi) and intralinguistic variety, in the different forms of English spoken by the characters. In the source version, English subtitles are provided every time Hindi or Punjabi is spoken. This switch between languages reflects contemporary middle and upper-class Indians, for whom, as Si (2011) has pointed out, English is seen as a prestigious language. In India it is used at different levels of government, in education (particularly higher education and technical training), and in print, electronic media, and the publishing sector.

We carried out a descriptive study in which the data from the corpus was collected and analysed qualitatively. To account for the varying presence of L3 in films, we used the classification of multilingualism by Bleichenbacher (2008a), who adapted for Hollywood films Petr Mares's taxonomy of representations of multilingual discourse for fictional, literary texts[1]. Bleichenbacher established the following classification to account for the varying presence of L3 in films (Bleichenbacher 2008a). Replacement strategies are a continuum that ranges from complete linguistic replacement of L3 to its full use in films:

- Elimination. In this strategy, "any speech that would have been in another language is completely replaced with an unmarked standard variety of the base language" (Bleichenbacher 2008a, p. 180). This means that there is no actual presence of L3 in the film. This has also been labelled as *homogenisation* by Carol O'Sullivan (2007, pp. 82–83), following Meir Sternberg's classification (Sternberg 1981, pp. 223–24).
- Signalisation. The L3 is explicitly named in the film as a metalinguistic comment (Bleichenbacher 2008a, pp. 180, 185).
- Evocation. In this strategy, "characters speak a variety of the base language that is characterised by interference", usually in the form of accent or code-switching (Bleichenbacher 2008a, p. 180).
- Partial presence. L3 is present through "linguistic landscapes" in the form of written signs (Bleichenbacher 2008a, p. 189) or in the form of unrealistic code-switching from L3 to L1, used for the purposes of the story (Bleichenbacher 2008a, p. 191).
- Full presence. L3 is the main language of the film, as is the case of Maya in *Apocalypto* (Mel Gibson 2006; Bleichenbacher 2008a, p. 195).

## 3. Results and Discussion

### 3.1. Representing Social Identity and Culture through Linguistic Variation in Monsoon Wedding

If we compare *Monsoon Wedding* (the original version) with *La boda del monzón* (the Spanish dubbed and subtitled versions) we can find a great difference in how languages have been used in the source text and the target texts. As indicated above, the ST film is a

mix of languages with English, Hindi, and Punjabi spoken at different times by different family members. Thus, English-speaking audiences can follow the film by listening to the English dialogues and reading the English subtitles provided for the Indian languages, while at the same time being able to appreciate the Indian accent of these middle-class citizens when they speak English. As Bonsignori and Bruti (2014) have put it, code-switching is also extensively employed in MW, where it has a social function. In the film, members of the upper-middle class mostly speak Indian English, sometimes switching to Indian languages, whereas Indian people from lower classes, such as the wedding planner Dubey, the workers, and the maid, always speak Hindi among themselves.

Notably, almost all the family members who gather at Verma's house for an arranged marriage in New Delhi can mix English and Indian languages, which represents modern Delhi. The exceptions are Aditi's grandmother and Dubey's workers, who only speak Indian languages, which echo ancient India. Dubey, the entrepreneurial wedding manager, speaks both English and Hindi to Verma's family, and Hindi to his colleagues.

Hence, in MW the use of the third language, together with some other cultural elements shown non-verbally, such as flowers, clothes, food, and decoration, as we will see in Section 3.3 below, enable Nair to depict the mixture of unique traditions of the past and modern touches in contemporary India. According to Geller (2002, p. 43), the film portrays the compounding or amalgamation of Indian tradition and Western modernity: "the film presents us with an India of computers and cell phones used by people who continue to live cultural life that values obligation, kinship, and custom above individuality". Likewise, Sharpe (2005, p. 59) states that:

> The audience witnesses Delhi street scenes of pushcarts and bicycle rickshaws weaving in and out of cars driving by a monolithic statue of Shiva. Golfers ride in golf carts across an immaculately landscaped golf course, while a row of women carrying sand in baskets on their heads (presumably for the sand pits) passes behind them. The camera often zooms in on television screens and monitors to emphasize the power of the new media, and it presents a TV talk show on film censorship, where guests debate the erosion of Indian morality and Hindu tradition. The heroine, Aditi (Vasundhara Das), represents a new generation of Indian women who live double lives in order to reconcile their desires with the wishes of their parents. Aditi secretly meets the man she loves the night before she is to marry the Houston NRI her parents have arranged for her to marry. (Sharpe 2005)

MW contains different languages in its source text to represent Indian cultural heritage and diverse social backgrounds. However, in the Spanish dubbed version almost all the characters speak the same language: Spanish; hence, L3 instances, including the Indian-accented English so common in the ST, can hardly be heard in the dubbed TT. The connotations of Indian-accented English in Britain or in the United States, where this accent is easily recognised, would probably be very different from those of Indian-accented Spanish in Spain, which can hardly be recognised.

The most complicated expressions to translate into another language and cultural background are usually those that are directly related to the source culture, i.e., cultural references, which "can be either exclusively or predominantly visual (an image of a local or national figure, a local dance, pet funerals, baby showers), exclusively verbal or else both visual and verbal in nature" (Chiaro 2008, p. 156). In MW some of the cultural references expressed verbally have not been transferred as such in *La boda del monzón*. Instead, they have been adapted to the TT culture. For example, Hemant wants to drink a "salt lassi" in the ST and "a tea" in the Spanish dubbed version (see Example, in Table 1 below).

**Table 1.** Salt Lassi.

| Example 1 | ST-English | TT-Spanish Dubbed Version |
|---|---|---|
| 0:31:27 | *Aditi:* Do you want something? <br> *Hemant:* I'll have a 'salt lassi', please. | *Aditi:* ¿Quieres tomar algo? [Do you want something?] <br> *Hemant:* Tomaré un té por favor. [I'll have a tea, please.] |

In example 1, the third language used to refer to "salt lassi", the traditional yoghurt-based drink from India, clearly signals otherness and makes the English-speaking audience aware of a foreign element although it might be recognised. In the Spanish version "salt lassi" has been changed into tea, a very different kind of drink, probably because the penetration of Indian culture, particularly culinary terms, is not so common in Spain as it might be in countries such as the United Kingdom or the United States. Consequently, the presence of L3 and its function to signal otherness have disappeared in the TT; when this happens, "multicultural identities and transcultural connotations are thus partially obscured" (Beseghi 2019).

On other occasions, cultural references have been completely omitted (see example 2 in Table 2 below). In this instance, the reference "sangeet" has not been transferred to the target text, and as a result otherness is not signalled here. Thus, according to Bleichen-bacher's classification, as shown in Section 2, the strategy used is that of "elimination", or "homogenisation", according to O'Sullivan (2007, pp. 82–83).

**Table 2.** Sangeet.

| Example 2 | ST-English | TT-Spanish Dubbed Version |
|---|---|---|
| 0:42:15 | *Varun*: Not more interruption. We'll never get this right. | *Varun*: No más interrupciones. Así nunca lo haremos bien. [No more interruptions. We'll never get this right.] |
| | *Ayesha*: What is it Rahul? We're rehearsing for the *sangeet*. What do you want? | *Ayesha*: ¿Qué pasa Raúl? Estamos ensayando. ¿Qué quieres? [What is it Rahul, we're rehearsing. What do you want?] |

In this scene, Ayesha and Varun are rehearsing a dance for the *sangeet*: a pre-wedding ceremony in Indian culture in which relatives and friends meet to celebrate the occasion. In the ST, Ayesha tells Rahul that they are rehearsing for the *sangeet*, but in the TT she simply tells him "estamos ensayando" (we are rehearsing), without mentioning what they are practising. The Indian word *sangeet* is not translated here, but elsewhere it is rendered in different ways, as shown in Table 8, in Section 3.2 below.

Hence, the strong presence of Indian cultural elements encountered in the ST is much weaker in the TT because (most of) the verbal references have disappeared and the audience can only enjoy the visual elements, of which there are many (see Section 3.3 below).

Table 3 below summarises how the languages of MW were mostly transferred into the dubbed version in Spanish, although there are a few exceptions, such as the cultural references *pakoras*, *samosa*, *chai*, *didi*, and *bhai-sahab*, that are kept in the TT.

**Table 3.** Languages in Monsoon Wedding/La boda del monzón.

| Monsoon Wedding | La boda del monzón |
|---|---|
| L1 | L2 |
| Indian English | Spanish |
| L3$^{ST}$ | L3$^{TT}$ |
| Hindi/Punjabi | Spanish (=L2) |

As can be inferred from Table 3, it does not matter whether the third language is a variety or an official language: it is mostly translated into standard Spanish in the dubbed version. The impact of linguistic varieties in a text depends as much on the ways they are embedded in the text as on the values and social representations attached to them. In MW, the linguistic variety takes place both through different languages (Indian English, Hindi, and Punjabi) and intralinguistic variety, in the different forms of English spoken by the characters. Some relatives attending the wedding are from Australia, and the groom is of Indian origin but lives in Texas and therefore is a Desi (a person who lives in the Indian subcontinent) who speaks this variety of English "resulting from the encounter/clash between English and South-Asian languages" (Bonsignori and Bruti 2014).

As mentioned before, linguistic variation is usually linked to social identity and culture, and it might be used in a text to highlight a particular aspect of society or to distinguish a group of people. As Newmark has written:

> Language is a substantial but partial reflection of a culture, culture being defined as the total range of activities and ideas and their material expression in objects and processes peculiar to a group of people, as well as their particular environment. (Newmark 1991, p. 74)

Deciding whether to preserve the third language, and thus a particular feature of identity or culture, or to eliminate any foreign traces in the TT might depend on the priorities and restrictions of the translation. However, the decision will have some impact on the representation of culture and so the visibility of foreignness and/or otherness in the TT (see Section 3.3 below).

*3.2. Third Language Strategies in Monsoon Wedding*

In this section, we address the presence of L3 in MW, which is seen as a continuum from complete absence to full presence of L3. We use the classification by Bleichenbacher (2008a) as explained in Section 2. We follow Bleichenbacher's invitation to extrapolate the findings of his analysis to multilingual films "outside the Hollywood mainstream" (Bleichenbacher 2008a, p. 194) and to see how these have been translated. Therefore, we examine to what extent this classification applies to MW, in the crossover between Hollywood and Bollywood films (see Section 2 above).

In signalisation, L3 is explicitly named in the film in the form of a metalinguistic comment. According to Bleichenbacher, this strategy is more common in written texts. In audiovisual texts, its overall impact on film reception will depend on the audience's attentiveness to such mentions of the language (Bleichenbacher 2008a, p. 185). We could argue that this is true of Hollywood films, where often L3 is somewhat artificially included to give a flavour of exoticism or otherness that is not always inherent to the film's nature. This is different from MW, where the presence of L3 is part and parcel of the film, and often assigned specific social and cultural features. This is the case, for example, in a TV programme discussion, in which one of the participants says the following (see Table 4, below):

**Table 4.** Speaking Hindi.

| Example 3 | ST- English | TT- Spanish Dubbed Version |
|---|---|---|
| 0:05:39 | *Mr. Bhatt:* You think just because **you wear handloom and speak in Hindi** . . . that you represent the common man? | *Mr. Bhatt:* ¿Qué se cree usted? ¿Que **porque lleva un sari y habla hindi** representa la mayoría de la población? [Do you think that just because you wear a sari and speak in Hindi you represent the majority of the population?] |

Spanish audiences of the dubbed version of the film are only aware that a TV discussion has taken place in English and Hindi because Hindi is mentioned. Spanish audiences of

the subtitled version will also be aware of this if they notice that when Hindi is used, the subtitles are in italics, which is the most common convention for signalling L3 in subtitling. This will make them aware of the code-switching between Hindi and English in the same sentence.

Let us now consider the following instance of signalisation in a family conversation with constant code-switching between English and Hindi, as shown in Table 5, below. After an elder member of the family praises the Rais (the wealthy American part of the family) as cultured, Lalit, the bride's father, ironically says the following in Hindi, subtitled in English in the original version:

**Table 5.** A little English.

| Example 4 | ST-English | TT-Spanish Dubbed Version |
|---|---|---|
| 0:23:07 | *Lalit:* Speak **a little English** and you become/a very cultured family. | *Lalit:* Cuando hablas **su idioma** te conviertes en una familia culta. [When you speak their language you become a cultured family.] |

As shown in example 4, the Spanish dubbed version only mentions "su idioma" (their language). The fact that the specific language (here English) is not mentioned is a dubbing convention, a kind of covert translation strategy used to hide specific references to foreign languages, which is opposed to the signalisation strategy.

This example leads us to consider the following strategy: *evocation*. According to Bleichenbacher, *evocation* of L3 in Hollywood films takes place through interference, especially as regards lexis (specific words in L3) and phonetics (accent), since these are relatively easier for actors to imitate if they do not know the L3 that is being represented. This strategy is very common in mainstream films in English, even though it has a major disadvantage: "it can fuel language ideologies according to which anybody who is not an L1 speaker of English is somewhat linguistically challenged" (Bleichenbacher 2008a, p. 186). Other solutions that are intended to portray more realistic visions are to cast multilingual actors (Bleichenbacher 2008a; Santamaria 2019), especially in diasporic or migration films (De Higes Andino et al. 2019).

In MW, evocation is the most common strategy for presenting L3 in the film. This is done mainly through the overall presence of Vernacular Indian English, and through code-switching between Hindi or Punjabi and English (see Section 1 above). This code-switching is presented as *part-subtitling*, which Carol O'Sullivan has defined as "a strategy for making a film shot in two or more languages accessible to viewers. [ . . . ] Part-subtitles, also called partial subtitles (O'Sullivan 2011), are appended to part of the dialogue only, are planned from an early stage in the film's production, and are aimed at the film's primary language audience" (O'Sullivan 2007, p. 81). In MW, English subtitles are used for Hindi/Punjabi exchanges in the original soundtrack, in scenes of code-switching with English. These L3 occurrences are dubbed into Italian (Bonsignori and Bruti 2014) or Spanish in their respective versions. Therefore, code-switching is not represented in the dubbed versions.

In the Spanish subtitled version, the part-subtitles of L3 in the original are presented in italics, in contrast with the rest of the subtitles, which are in Roman type to render English dialogues. Since Spanish audiences generally cannot distinguish between Hindi and Punjabi, this use of italics could be considered an extradiegetic form of signalisation.

Contrary to the L3 presence in Hollywood films only through lexis and phonetics, in Bollywoodian films morphosyntactic features of Indian English are present. In translation, these features are often neutralised, as in the Italian translation of MW analysed by Bonsignori and Bruti (2014) and in the Spanish dubbed version. Therefore, in practical terms, L3 is rendered in translation mostly through the introduction of loan words from Hindi or Punjabi from the domains of food and cooking, as terms of address and greetings, or words related to the wedding ceremony, which create an effect of code-mixing (Bonsignori and Bruti 2014). For Monti (2019, p. 225), the "recurrent naming of food in the ethnic char-

acters' discourse practices reinforces the role multicultural films play in communicating transcultural stereotypes as well as transnational linguistic identities" because consuming a certain type of food is representative of a particular socio-cultural tradition. Some food examples are shown in Table 6 below.

**Table 6.** Food and cooking terms.

|  | ST-Hindi | TT-Spanish Dubbed Version | TT-Spanish Subtitled Version |
|---|---|---|---|
| 00:02:01 | nimbu pani | limonada [lemonade] | nimbu pani |
| 00:02:28 | pakoras | pakoras | pakoras |
| 00:31:24 | salt lassi | té [tea] | salt lassi |
| 00:39:42 | chuski | helado [ice cream] | Polo [ice pop] |
| 00:49:35–55 | samosa | samosa | dulce [sweet] |
| 01:02:59 | chai | chai | chai |

In the translations into Spanish, quite erratically, as pointed out in Minutella (2012, p. 317) for the cases studied into Italian, different domesticating or foreignising choices coexist in the dubbed and subtitled versions. In the dubbed version of MW, half of the referents (*pakoras, samosa*, and *chai*) are preserved, whereas the rest have been adapted. This contrasts with the Italian translation of MW in which *lassi* and *samosa* are preserved, whereas the rest have been adapted (Bonsignori and Bruti 2014).

There is more variation in the terms of address and greetings, as shown in the examples presented in Table 7 below:

**Table 7.** Terms of address and greetings.

|  | ST-Hindi | TT-Spanish Dubbed Version | TT-Spanish Subtitled Version |
|---|---|---|---|
| 00:04:28 |  | hijo [son] | Ø |
| 00:04:37 |  | hijo [son] | Ø |
| 00:54:03 |  | hijo [son] | Ø |
| 01:08:00 | beta | Ø | Ø |
| 01:08:40 |  | hijo [son] | Ø |
| 01:08:58 |  | hijo [son] | Ø |
| 00:08:43 | didi | didi | Ø |
| 00:42:09 |  | didi | Ø |
| 00:49:38 |  | Ø | Ø |
| 00:50:05 |  | didi | Ø |
| 00:17:17 | bhai-sahab | bhai-sahab | Ø |
| 00:20:08 |  | bhai-sahab | Ø |
| 00:20:10 |  | bhai-sahab | Ø |
| 00:23:13 |  | Ø | Ø |
| 00:26:38 |  | bhai-sahab | Ø |
| 01:07:32 |  | Ø | Ø |
| 01:33:16 |  | bhai-sahab | Ø |
| 01:33:42 |  | bhai-sahab | Ø |
| 00:20:14 | bhai | bhai | Ø |
| 01:37:32 |  | hermano [brother] | Ø |
| 00:17:43 | namaste | namaste | Ø |

*Beta* is mostly translated as "hijo" (son), whereas the respectful forms of address for women *didi* or for men *bhai-sahab* are mostly kept in dubbing, in contrast with the Italian dubbed version, where they are omitted in half of the examples analysed (Bonsignori and Bruti 2014). The respectful salutation *namaste* has been kept once in dubbing, but most often is shown visually and not verbally (see Section 3.3 below).

Finally, the terms connected to the wedding ceremony are variously rendered as shown in Table 8:

**Table 8.** Celebration terms.

|  | ST-Hindi | TT-Spanish Dubbed Version | TT-Spanish Subtitled Version |
|---|---|---|---|
| 00:24:17 |  | sangeet | sangeet |
| 00:42:20 |  |  | sangeet |
| 01:02:00 | sangeet | sangeet | sangeet |
| 01:15:25 |  | unión [union] | boda |
| 00:20:38 | shadi | boda [wedding] | boda |
| 01:07:54 |  | sangeet | sanget |
| 00:49:26 | mehndi | dibujo [drawing] | henna |
| 01:08:29 | dupatta | dupatta | dupatta |

Thus, *sangeet* is maintained or adapted in the same film, as is the case with the Italian translation. Curiously, *shadi* has been adapted once as "boda" (wedding), but also replaced by *sangeet* once. As regards *dupatta*, the loan-word has been kept, in contrast to the Italian version, where it has been adapted as "velo" (veil), in keeping with the adaptation of *mehndi*, both in Spanish and Italian. We agree with Bonsignori and Bruti (2014) about the probable reasons for keeping loan words in the Spanish and Italian dubbed versions of MW: "the relative scarcity of culturally imbued words referring to the exotic artefacts or habits does not impair comprehension and might be welcomed by the audience". However, code-switching is eliminated, which Bonsignori and Bruti (2014) consider "linguistic whitewashing".

L3 can also take the form of *linguistic landscapes* (Bleichenbacher 2008a, pp. 189–91), that is, the presence of L3 as written signs. Bleichenbacher refers mainly to written signs in English in Hollywood films. In MW, linguistic landscapes take the form of written signs in English or Hindi in scenes outdoors showing the busy life in New Delhi.

In this section, we have analysed Bleichenbacher's strategies, which are mostly verbal, probably because they are an adaptation of Mares' taxonomy for written fiction. From our previous research (Corrius 2008, and TRAFILM and MUFiTAVi projects), we argue that the presence of L3 can also be represented non-verbally, as analysed just below (Section 3.3).

*3.3. Otherness in Monsoon Wedding*

An increasing number of films use L3 to recreate the multicultural present world. Nevertheless, multilingualism delivers a feeling of otherness. As explained in the first section of this article, code-switching in Indian films generates an authentic atmosphere, a feasible representation of daily life in India. However, audiences without knowledge of Punjabi and Hindi may feel the gap generated by the use of these languages. For this reason, we aim to investigate what happens when the film is translated into Peninsular Spanish.

For audiences outside India, otherness is present in visual and aural channels. In the visual channel, the images of the streets of India help to produce authenticity because they seem to be taken from a documentary, as explained above. Cultural referents are also relevant, both when they appear in the images and when they are included in the script (see Section 3.2). We can identify a third area where otherness plays a major role in MW, since one of its topics is the clash between tradition and modernity. Hence, social groups are shown sharing different values and ideological representations. From the very beginning of the film, the audience can see the preparations for a traditional Indian wedding, so they are presented with an image of traditional India. Nevertheless, as we get to know more about the characters, we realise they are not only defined by features that can be univocally associated with a given group. Thus, Dubey's mother, who tends to complain about her son's singleness, is nevertheless interested in the ups and downs of the stock

exchange. She is portrayed as someone at the junction between the traditional and the new and prosperous India. One of the main characters, Aditi, decides to agree to marry a man she barely knows after her relationship with her married boss did not work out the way she expected. Consequently, Aditi is portrayed as a young woman who tries to find her place in a society that is undergoing rapid value changes. Her cousin Ria, who challenges Aditi's decision and has a strong attitude against arranged marriages, confronts her by saying: "So what do you do? Get married to some guy selected by mummy and daddy, you've barely known him for a couple of weeks!" As a result, even the clash between tradition and modernity creates a perception of "otherness" at different levels since the characters shift between old and new values.

Mira Nair conveys the concept of "otherness" within India through various details in the script and the images. However, Spanish viewers may not be able to track the meaning of the social representations shown in MW if they do not have enough previous knowledge of Indian culture. In particular, Spanish audiences may tend to add new features of otherness to the behaviour of certain characters, as we will show below with the example of marigolds.

Marigolds, which are very popular in India and have become the flowers of religious ceremonies, especially in the Sikh tradition, appear sixteen times in the film but the name of the flower is never verbalised. They appear from the very first moment when we see them falling from the wedding decorations. Marigolds are a symbol that is used so much in MW that the Spanish audience might finally perceive its related values, although it is hard to establish to what extent or how much they will differ from the meaning understood by the Indian audience. The Spanish audience may be even more bewildered when they see Dubey eating them. Therefore, there is a real chance that the character of Dubey will be interpreted inadequately.

According to Moscovici (1984, p. 27), social representations can be learned because they have as an intrinsic characteristic the aim to "make something unfamiliar, or unfamiliarity itself, familiar". Therefore, Moscovici considers that new social representations can be acquired through two cognitive processes: anchoring and objectification. By anchoring (Moscovici 1984, p. 29), inexplicable viewpoints or new concepts are transformed into previously learned ordinary categories. Subsequently, individuals do their best to objectify them by turning "something abstract into something almost concrete, to transfer what is in the mind to something existing in the physical world" (Moscovici 1984, p. 32). This is how we learn about others and their social representations. MW was one of the first Indian films to become a box-office success in Europe (Binimelis-Adell and Saavedra Montaño 2005). Therefore, the general public with no previous knowledge of the cultural referent of marigolds was probably puzzled by the repeated display of this flower throughout the film, while at the same time trying to uncover the information this flower could carry. Then again, since Bollywood films have found an audience in Spain, Indian social representations could now be better understood.

Usually when we refer to cultural referents and media translation, we discuss mainly aspects that appear in the script. However, cultural referents are present on the screen as we have explained in the case of marigolds, comprising a given visual landscape. In a previous study (Santamaria 2017), the importance of the image was discussed. According to Charaudeau (2005, p. 188), the image fulfils three functions in cinematographic productions: *designation*, *figuration*, and *visualisation*. Through designation, the intention is to demonstrate that the reality of the audiovisual production is authentic. As has been underlined before, the aim of the street images in MW is to convey a flavour of the Indian atmosphere to the viewer. In the film, the images of streets full of active people, with chaotic traffic and sudden rain, trigger a certain common stereotype in Spain that is confirmed by the quality of the images in MW, which seem to be taken from a documentary. Thus, we interpret that Nair is attempting to make us realise how the story she is creating fits in a realistic portrait of India.

Charaudeau (2005) defines figuration as the reconstruction of an existing world, which is not a copy of reality, but rather a construction-representation of something that is already known, which aims to create a sense of "credibility". There are various cultural referents in the images and music that are not going to be easily understood by an audience with little past contact with India. We focus on some of the referents in the images. Again, as in the case of marigolds, the public has to try to make sense of the greeting of *namaste* or the different sizes, patterns, and colours of the *bindi* (probably without being able to associate them with personal preferences or genre, age, social status, etc.). In the same way, in the first minutes of the film we can see the magazine *Cosmopolitan* that the public may or may not identify with a given content and the values and social representations associated with it. The Spanish audience may find it challenging to associate the various decorations, drinks, food, and clothing with the different steps of wedding parties. However, the social representations of the cooking classes on TV or the overdose of perfume by C.L. Shandha are easier to figure out.

Finally, for Charaudeau (2005), visualisation is representation through a code system, a way of organising the world that aims to recreate the effect of "discovering the truth". Within this category, it should be noted how the audience of MW gradually discovers one of the most relevant themes of the film, child abuse. The public learns through the images, and before any conversation is held about this main topic of the film, about Udai Verma and how his misbehaviour affected Ria's life. In the same way, we learn how Dubey and Alice fall in love without having to hear them pronouncing any words, or why Pimmi hides herself in the bathroom to smoke. Previous knowledge helps to frame meaning.

Consequently, one of the central topics in MW, otherness, may not be properly understood in Spain since complete decoding of the meaning of the images is only possible if the social representations are properly recognised. The translation into Peninsular Spanish should take into consideration which images require an explanation, whenever possible.

The manifold presence of otherness in MW will acquire new meanings in the translation because the implications will be perceived in a different way by the Spanish audience. The images will be understood according to viewers' previous stereotypes of India, and the cultural referents in the images and in the script will prompt the audience to add new knowledge to their previous categorisations of social representations. It does not help that, as mentioned above (Sections 3.1 and 3.2), cultural referents are sometimes fully domesticated (as in the case of "chuski") and in other instances they are totally foreignised (as with the title of veneration "bhai-sahab").

## 4. Conclusions

From our research in our previous TRAFILM and MUFiTAVi projects, we can state that L3 is used to recreate specific worlds, representing a given multilingual culture. Languages mark geographical and cultural borders and embody the characters' social, cultural, and personal features. Hence, linguistic variation is usually linked to social identity and culture, and it might be used in an audiovisual text to highlight a particular social aspect, or to distinguish a group of people.

Thus, multilingualism in Bollywood productions, and specifically in *Monsoon Wedding*, reflects Indian society, in which code-switching between English and Indian languages is common. The linguistic variety of the film includes Indian English, Hindi, and Punjabi together with intralinguistic varieties, mainly the different forms of English used by the characters. L3 becomes a tool for Nair to reveal the essence of the Indian spirit: unique traditions of the past combined with modern touches.

Language variation in source texts may be transferred in three different ways: (a) keeping the same L3 language as in the ST, (b) standardising the ST language in the target text as part of its main language, and (c) selecting a language different from L1, L3ST, or L2. In *Monsoon Wedding*, the Spanish dubbed version essentially uses the second choice, in that the third language disappears in the target text. However, on occasion, the third language is maintained when it refers to culture elements such as names of traditions (e.g., *sangeet*),

food terms (e.g., *nimbu pani*), and terms of address or greetings (e.g., *namaste* and *bhai-sahab*). In the subtitled version, only the terms of address and greetings are omitted.

The strong presence of Indian cultural elements encountered in the ST is much weaker in the TT because (most of) the verbal references have disappeared, and the audience can only enjoy the visual elements.

Following Bleichenbacher's suggestion to test the validity of his taxonomy for films outside Hollywood (Bleichenbacher 2008a, 2008b) and to include translations, we found that signalisation has scarcely been used in dubbing. However, the use of italics in subtitles might be considered a kind of signalisation strategy to show the presence of L3. Evocation is the most common strategy that is found. Code-switching in Bollywood films shows the speakers' linguistic behaviour in multilingual India and is far more common than in Hollywood films. Code-switching in *Monsoon Wedding* goes beyond lexis and phonetics and includes morphosyntactic aspects of L3. These are partially obliterated in the Spanish dubbed version.

When L3 is not rendered in translation, what remains is image and music, although these non-verbal elements may not be always understood by audiences that are not familiar with Indian culture. Concerning the verbal aspects of the film, when the audience of the translated version does not understand the values and social representations associated with the cultural elements in the original language, the message may not get through and the public may be led to multiple misunderstandings. Nevertheless, the otherness in *Monsoon Wedding* does come across, and Spanish audiences can perceive the foreignness portrayed in the original version, as proved by the interest generated in Bollywood films among the Spanish public.

**Author Contributions:** Conceptualization, M.C., E.E. and L.S.; methodology, M.C., E.E. and L.S.; investigation, M.C., E.E. and L.S.; resources, M.C., E.E. and L.S.; writing—original draft preparation, M.C., E.E. and L.S.; writing—review and editing, M.C., E.E. and L.S. All authors have read and agreed to the published version of the manuscript.

**Funding:** This research received funding from the following two projects: MUFiTAVi (2019–2021) Multilingualism in audiovisual fiction and its translations for Spain in digital platforms (PGC2018-099823-B-I00), funded by the Spanish Ministry of Science and Innovation. TRAFILM (2015–2018) The Translation of Multilingual Films in Spain (FFI2014-55952-P) funded by the Spanish Ministry of Economy and Competitiveness.

**Institutional Review Board Statement:** Not aplicable.

**Informed Consent Statement:** Not aplicable.

**Conflicts of Interest:** The authors declare no conflict of interest.

## Note

[1] Mareš, Petr. "Fikce, konvence a realita: K vícejazycnosti v umeleckých textech (Fiction, convention, and reality: On multilingualism in literary texts)." In Bleichenbacher (2008a, pp. 180–81).

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

## Research Projects

