# Peer review of "Translating Multilingualism in Mira Nair’s Monsoon Wedding"

_languages, doi:10.3390/languages8020129_

Reviewer 1 Report
A gloss/back-translation in English should be added sistematically to all the examples provided in Spanish.
You refer to a quantitative analysis of the data, but you reduce it to some tables without examining them in any great detail.
Hindi & Punjabi are considered together as L3. Refer to Corrius and Zabalbeascoa (2011, 2019) once more to understand where the problem lies.
Reference to Monti (2009, 2014, 2016), Minutella (2012) and Beseghi (2016) might improve your overall argumentation.
You should deepen the issues discussed in the second part of the paper, which is more promisimg than the first one, That should be the way to go.
The overall impression is that your article comprises two different papers within, with neither being explored in depth as deserved.
Some parts of your text read as anecdotal: you do not provide any evidence or source for what is being stated. That is not a good academic practice.
Author Response
Dear Reviewer 1,
Thank you very much for your time, suggestions and comments, which have certainly helped us refine our manuscript entitled “Translating Multilingualism in Mira Nair’s Monsoon Wedding”.
We have paid careful attention to the details and recommendations. Attached, you can see your suggestions and our responses. However, if something needs clarifying or you need further information, don’t hesitate to contact us.
Best regards,

Reviewer 2 Report
Dear author(s), thanks for this interesting paper. Overall, I think it is good, but still there are a few things that could be improved in my opinion. I’ll expand on this below.
Theoretical framework section:
Bollywood is not “your average mainstream” genre, but a specific genre also attracting a specific audience. Similar content, such as anime immediately springs to mind, of which the audience has a specific interest (more so than in mainstream content) in the ST culture. A certain openness to alternative translation strategies that allow for more of the ST (culture) to shine through in the translation can be expected. Of course, in this regard we also think of Nornes’s research on “abusive subtitles” (or “abusive AVT”) which could definitely be reflected upon in this contribution.
Methodology section:
What was unclear to me was: is the translation of ALL cultural references in the ST systematically studied? You also refer to the Italian AVT of the same film (Bonsignori&Bruti), was the Spanish version systematically analysed vis-à-vis the Italian AVT? See line 323. I believe this should be clarified better also was their methodology applied to this study?
Section 3.3 contains a very interesting discussion of the marigolds in the visuals. I believe this should be dealt with separately, rather than being added to this general “Otherness in Monsson Wedding” section. Please, consider using separate (sub)sections.
Also, I think we as AVT researchers need to be critical of certain common AVT practices rather than simply referring to what is “the done thing”, we can (and should) challenge certain AVT norms. Just like Nornes did by the way. From my own experience working with dubbing directors and subtitlers, I also notice that they are open to our views on what can be improved. A lot of the choices they make are not that well-considered and just build on what they believe(!) is common practice. Using foreignizing strategies particularly in this kind of films is perfectly justifiable also for Spanish-speaking audiences even if they are not expected to be familiar with the ST culture. Moreover, I believe the line between either dubbing or subtitling should not be drawn this radically: e.g. just like in the source text subtitles were added to make the Indian languages accessible to English speakers, the dubbed version could also include partial subtitles to convey these languages.
General comments
Reference to Mares’s work is missing in-text and in the reference list
Please provide back translations for all Spanish TT also in your tables
Wording:
- What do you mean by the greeting “namaste” being visually shown in line 336?
- What do you mean by “the real India” in line 52
Language issues: 
have the text proofread, please, see issues such as 
-commonest -> most common in line 293
-odd use of 'still' in line 365
-bracket missing in line 96, space missing in line 140
Author Response
Dear Reviewer 2,
Thank you very much for your time, suggestions and comments, which have certainly helped us refine our manuscript entitled “Translating Multilingualism in Mira Nair’s Monsoon Wedding”.
We have paid careful attention to the details and recommendations. Below, you can see your suggestions and our responses. However, if something needs clarifying or you need further information, don’t hesitate to contact us.
Best regards,

Round 2
Reviewer 1 Report
Page 12:
“MW is one of the first Indian films to become a box-office success in Spain”. This is anecdotical. Authors should refer to an official source for this statement to have academic value.
If they add a reliable source for this, no other remarks/complaints.
Author Response
Dear Reviewer,
We have included the academic reference as required.
Thank you so much for your comments and time.
Best regards,
